# A 241-Year *Cryptomeria fortune* Tree-Ring Chronology in Humid Subtropical China and Its Linkages with the Pacific Decadal Oscillation

**Zhipeng Dong [1], Dai Chen [2], Jianhua Du [3], Guang Yang [4],\*, Maowei Bai [1], Feifei Zhou [1], Zhuangpeng Zheng [1], Chaoyue Ruan [1] and Keyan Fang [1],\***

[1] Key Laboratory of Humid Subtropical Eco-geographical Process (Ministry of Education), College of Geographical Sciences, Fujian Normal University, Fuzhou 350007, China; dongzhipeng170@163.com (Z.D.); 17863349965@163.com (M.B.); chenqy_gogo@126.com (F.Z.); 13163892962@163.com (Z.Z.); chaoyueruan@sina.com (C.R.)

[2] National Forestry and Grassland Administration (National Park Administration), Beijing 100714, China; treering_chendai@126.com

[3] Forest Fire Prevention and Monitoring Information Center of the State Forestry and Grassland Administration, Beijing 100714, China; du_jian_hua@yeah.net

[4] Key Laboratory of Sustainable Forest Ecosystem Management (Ministry of Education), School of Forestry, Northeast Forestry University, Harbin 150040, China

\* Correspondence: yangguang@nefu.edu.cn (G.Y.); kfang@fjnu.edu.cn (K.F.)

**Abstract:** Humid subtropical China is an "oasis" relative to other dry subtropics of the world due to the prevailing of the East Asian summer monsoon (EASM). Although many long climate sensitive tree-rings have been published to understand the historical climate change over various regions in China, long tree-ring chronologies in humid subtropical China are rare due to the difficulty to find old growth trees. This study established a tree-ring chronology spanning from 1776 to 2016 from *Cryptomeria fortunei* Hooibrenk ex Otto et Dietr in Liancheng area of humid subtropical China, which is also currently the longest chronology in Fujian province. Similar to the climate-growth relationships in neighboring regions, our tree-ring chronology is limited by cold temperature in winter and spring and drought in summer. In addition, a drought stress before the growing season also played a role in limiting the growth of our tree rings. Our climate sensitive tree rings showed different correlations with the Pacific Decadal Oscillation (PDO) in different periods, possibly via modulation of the EASM.

**Keywords:** tree ring; climate change; southeast China; PDO; *Cryptomeria fortunei*

## 1. Introduction

Understanding the effect of regional climate variability on forest ecosystems in the past can shed light on how current climate change impacts will affect forests in the future [1–4]. As the strongest component of the global monsoon, the East Asian summer monsoon (EASM) brings a high amount of the precipitation to the subtropics of China, home of the world's largest subtropical evergreen broad-leaved forest, known as the "oasis" on the tropic of cancer [5]. Various studies have been conducted on climate change and its impact on the forest dynamics of this area [2]. However, due to the relatively limited number of climate proxies from this region there are few paleoclimate studies able to place the recent climate change and forest dynamics in any significant, long-term perspective. Tree-rings are the most widely used proxy for climate reconstruction of the past millennium due their accurate dating, high climate sensitivity and high spatiotemporal resolution [6–8]. On the other hand,

tree-rings are also an important indicator of tree growth and thus forest dynamics, which is an ideal proxy to study the impacts of climate on forest dynamics [9–11].

Southeast China is a key region in the humid subtropical China and in recent years an increasing number of tree-ring studies have been carried out there [12–21]. These studies have shown that tree growth in southeastern China is often limited by the summer drought and winter coldness (dry) [15–20]. The climate sensitive tree rings in this core areas of the EASM are closely modulated by oceanic and atmospheric patterns, such as the El Niño-Southern Oscillation (ENSO), and Pacific Decadal Oscillation (PDO) [16,19–22]. Compared to the dense and long tree-ring chronologies developed in western and northern China [9–11,23–26], the tree-ring chronologies in southeastern China are still sparse, and many are shorter than 200 years. Here, we use a new tree-ring chronology from *Cryptomeria fortune* to investigate the climate-growth relationships and the climate history in the Liancheng region of western Fujian province.

## 2. Materials and Methods

### 2.1. Study Region and Tree-Ring Data

The sampling site (LY-Laiyuan) is located in the Laiyuan Town, Liancheng County, Fujian Province (Figure 1). Laiyuan Town is a traditional Hakka village (moved here from the central plains during the Ming and Qing dynasties), located at the north foot of the Meihua Mountain. The annual mean temperature of the study region is 19.8 °C, and the annual mean total precipitation is about 1600 mm. The study area is warm during April–October with the peak in July (Figure 2a), while it is relatively dry during the hot season (July to October) due to the influence of the subtropical anticyclone over the western pacific in summer. Precipitation is concentrated in March–June and peaks in May and June. During the instrumental period (1957–2016), the annual mean temperature exhibited a statistically significant increasing trend after the late 1980s (Figure 2b), while the total precipitation showed an obscure increasing trend (Figure 2c). The tree-ring samples of *Cryptomeria fortunei* were taken from patches of old growth trees in a natural forest and few planted trees in urban settings. *Cryptomeria fortunei* is an evergreen coniferous species, widely distributed in humid subtropical region from the southernYangtze river basin to Guangdong, Guangxi and Guizhou provinces in southern China. It generally grows in the mountains where it is warm and humid and cooler in summer [27]. *Cryptomeria fortunei* trees often co-exist with the *Cunninghamia lanceolata* and other dense shrubs and bamboos.

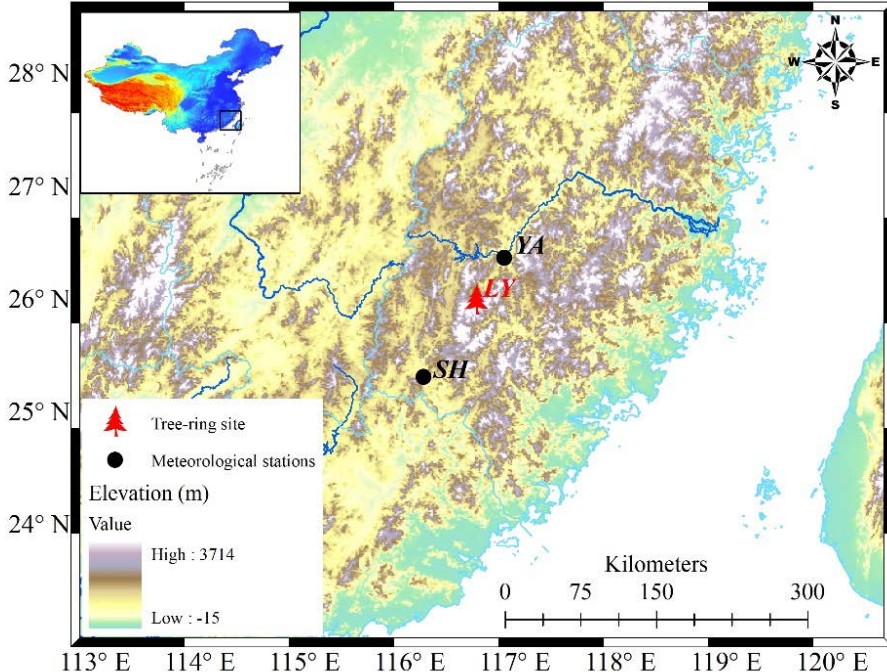

**Figure 1.** Locations of the sampling site (LY—Laiyuan) and meteorological stations (YA—Yong'an; SH—Shanghang).

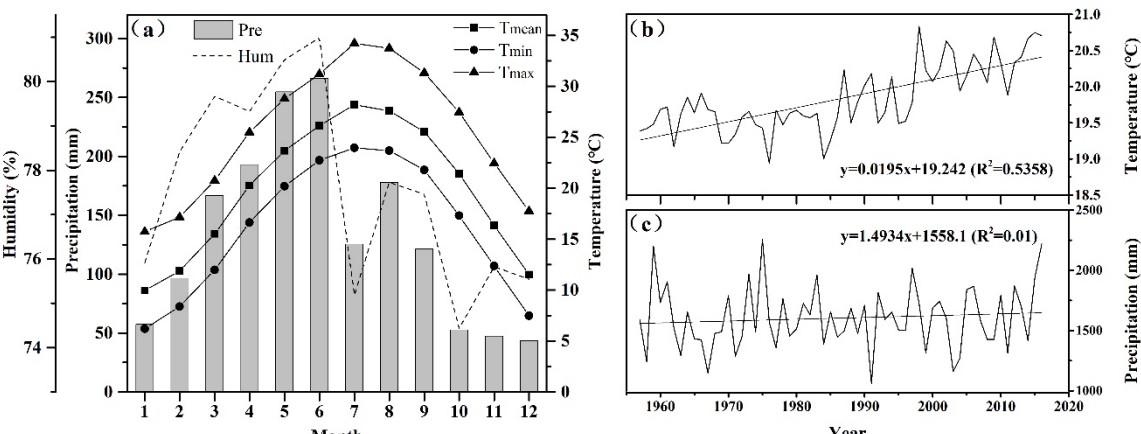

**Figure 2.** Climatic records from the Yong'an meteorological station and Shanghang meteorological station. (**a**) Monthly maximum (Tmax), mean (Tmean), minimum (Tmin) temperatures, humidity (Hum) and precipitation (Pre); (**b**) multi-year averaged annual mean temperature and (**c**) mean annual total precipitation. Straight lines indicate the linear trends.

We collected 82.5 mm diameter cores at breast height from 30 healthy *Cryptomeria fortunei* trees at locations from 950–1000 m above sea level, on slopes from 10 to 25 degrees. All samples were treated according to the standard dendrochronological methods [28], including air-drying, mounting and polishing with progressively finer sand paper until the surfaces were smooth and the ring boundaries were clear and distinguishable under a microscope. Visual crossdating was used to preliminarily assign the calendar years to each growth ring. The width of each annual ring from all the cores was measured using the LINTAB-6 measurement machine with precision up to 0.001 mm. The quality of crossdating was checked using the COFECHA program [29]. The longest series from LY extends from 1688 to 2016 AD and the ratio of missing rings is 0.032%. The cross-dated series were then standardized in program ARSTAN using smoothing splines with a 50% frequency-response cutoff of two-thirds of

the series length to remove the age-related growth trends [30]. Finally, the de-trended tree-ring series were averaged, by bi-weight robust-mean estimation, to generate a standard (STD) chronology [30]. The subsample signal strength (SSS) over 0.85 was used to evaluate the reliability of the tree-ring chronologies [31]. To further evaluate the quality of the ring-width chronology, the running correlation (Rbar) and the expressed population signal (EPS) were calculated using a 50-year window with 25-year overlap [28,31] from 1917–2016, the common interval among all samples.

## 2.2. Climate Data and Analytical Methods

Monthly total precipitation, mean temperature and relative humidity were extracted from the two nearest Yong'an (25°58′ N, 117°21′ E, 206 m a.s.l.) and Shanghang (25°03′ N; 116°25′ E, 198 m a.s.l.) meteorological stations, which are 50 km northeast and 80 km southwest of the study site LY (Figure 1). We calculated the mean climate data from the two stations to represent the regional climate, as records from the two stations were highly correlated. The correlation coefficient (r) between the station's annual mean temperature, total precipitation and humidity are 0.94, 0.70 and 0.74, respectively.

We additionally employed the Standardized Precipitation and Evapotranspiration Index (SPEI) [32] from the nearby grid points (25°45′ N, 117°15′ E, 1957–2014). SPEI measures anomalies in climatic water balance (precipitation minus potential evapotranspiration) from 1901 to 2014 at a spatial resolution of $0.5° \times 0.5°$. Compared with other drought indices, SPEI can produce a drought index on multiple timescales [32]. We extracted the SPEI data from 1957 to 2014 for analysis to conform to the time span available from the nearest meteorological records.

We compared our tree ring chronology with the HadISST1 sea-surface temperature (SST) dataset [33]. The correlations with SST were conducted using the KNMI Climate Explorer (http://climexp.knmi.nl) [34]. To analyze the Climate-growth relationship the Pearson correlation between the mean annual climate variables of temperature, precipitation, and SPEI were calculated for 16 months from previous August to current November; the late growing season of the previous year to the end of the current growing season.

## 3. Results

### 3.1. Tree-Ring Chronology

The final STD chronology consisted of 61 tree-ring series from 22 trees. This is after discarding those series that were not datable or too short. The reliable period of the tree-ring chronology is from 1766 to 2016 (Figure 3). Table 1 shows the statistics of the chronology, the mean sensitivity (MS) and standard deviations (SD) of the chronology are 0.314 and 0.281, respectively, which are similar to other tree-ring chronologies in southeast China [13,18]. A high MS indicates high variability between adjacent ring widths, which is likely related to climate variability. The first-order autocorrelation (AC1) of the chronology is 0.739, indicating a high degree of association between successive measurements.

The Signal to Noise ratio (SNR) and the explained variance are 7.004 and 20.1%. These values are not high relative to the arid and cold regions, likely due to strong local disturbances that often occur in dense forests of humid and hot regions [17].

**Table 1.** Statistics of the tree-ring index and common period (1917–2016 AD) analysis.

| | |
|---|---|
| Mean Sensitivity (MS) | 0.314 |
| Standard deviation (SD) | 0.281 |
| First-order autocorrelation (AR1) | 0.739 |
| Expressed population signal (EPS) | 0.875 |
| Signal-to-noise ratio (SNR) | 7.004 |
| Variance in first eigenvector (PC1) | 20.1% |
| First year where SSS > 0.85 (tree number) | 1776 (5) |

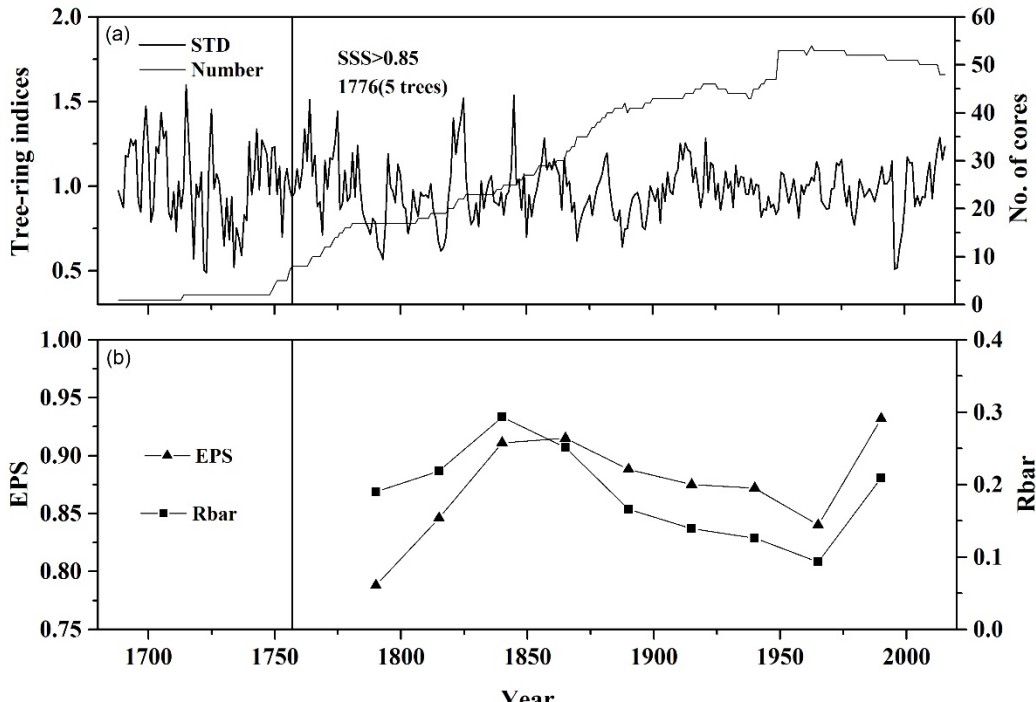

**Figure 3.** (**a**) The tree-ring STD chronology of the Laiyuan Town and the number of cores, (**b**) Running Rbar (running based upon a 50 yr window with a 25 yr lag) and statistics of the running expressed population signal (EPS).

### 3.2. Climate-Growth Relationship

As shown in Figure 4, the tree rings are positively affected by the spring, autumn and winter temperature. For example, significant ($p < 0.05$), positive correlations are found between the chronology with mean temperature in current February (0.27) and September (0.27), with minimum temperature in current February (0.28) and May (0.27), with maximum temperature in current September (0.31), as well as seasonally averaged temperature from current February to September (Figure 4a). For the first differenced data (Figure 4c), significant ($p < 0.01$), positive correlation is found between the chronology with minimum temperature from previous December to current April (0.36). The radial growth of these trees is negatively affected in summer temperature. For example, significant ($p < 0.05$) negative correlation was found between the chronology with the average temperature (−0.27) and maximum temperature (−0.34) in current June (Figure 4c).

There are positive correlations between tree growth and summer precipitation, suggesting a summer drought stress (Figure 4b,d). In addition, positive correlations were also observed with previous winter precipitation (Figure 4b,d). Positive correlations with seasonally averaged precipitation (0.40) and humidity (0.32) are seen from previous November to December. For the first differenced data, the correlation values between the chronology with the precipitation and humidity in current June are more significant than the correlations with raw data (Figure 4b,d).

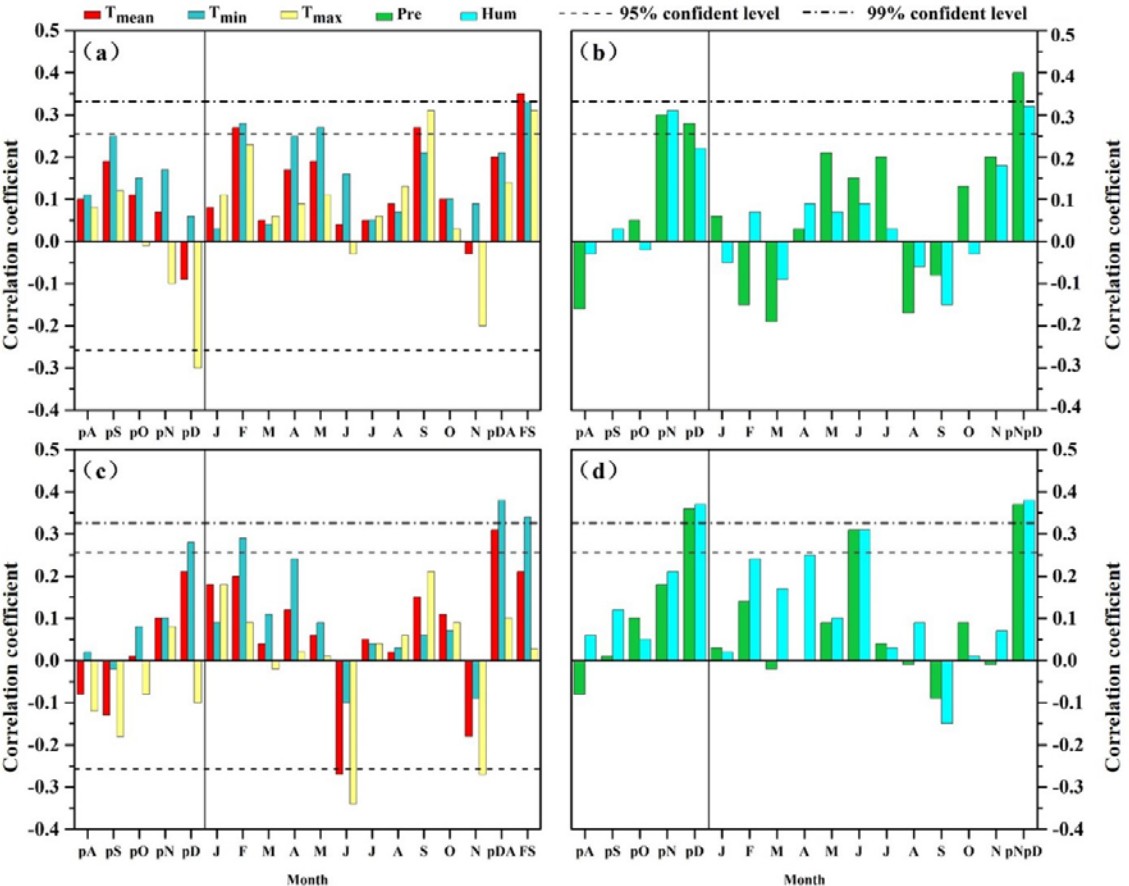

**Figure 4.** Correlation coefficients between the tree-ring chronology of LY and the regional climate. (**a**,**b**): STD; (**c**,**d**): First differenced data. p: previous year; pDA: previous December–current April; FS: February-September; pNpD: previous November–previous December.

Above analysis indicated both the drought in summer of current year and cold temperatures and drought in winter of previous year are the limiting factors for tree growth of this area. High temperature and precipitation in winter of previous year are more important for tree growth than hot and rainy summers in the current year as indicated by the more significant climate-growth correlations. We further investigated the responses of tree rings to SPEI on different timescales (Figure 5). The chronology has more significant correlations with SPEI on short timescales, from 1–3 months, than long timescales, from 4 months and 6–11 months. Similarly, the chronology has more significant positive correlations with SPEI in winter of the previous year than in summer of the current year (Figure 5a). On the other hand, when first-differenced, the chronology has significant positive correlations with different monthly SPEI in the summer of the current year (Figure 5b).

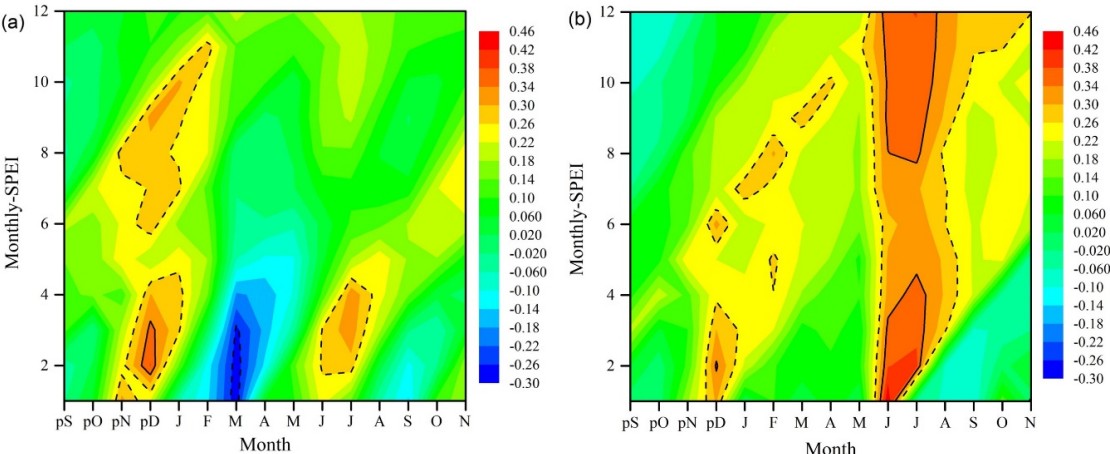

**Figure 5.** Correlations calculated between the indexed tree-ring width chronology and the SPEI drought index at monthly scales ranging from 1 to 12 months from 1957 to 2014. (**a**): STD, (**b**): First differenced data. The dash lines represent the statistical significance level of 0.05, continuous lines represent the statistical significance level of 0.01.

## 4. Discussion

### 4.1. The Climate Limiting Factor for Tree Growth

The radial growth of coniferous tree rings in southeastern China, such as the lower reaches of the Yangtze River basin and the Fujian Province, is widely seen to be limited by summer drought and winter coldness [12–16,20]. Tree growth is subjected not only to the influence of current year's climate, but also the previous year's [35]. The earlywood formed during the early period of the growth season partly uses stored photosynthate from the previous year [36]. Although radial growth stops between November and December, photosynthesis in this warm subtropical area may continue to produce carbohydrates that are useful for the growth of in trees the next year [37,38]. Low temperatures in winter and spring can reduce the metabolic activity in trees. In the case of extreme low temperatures, it can even cause damage to the roots, shoots or branches of the tree, affecting a low ring width in the following growing season [12,39,40]. On the other hand, cold winter and spring temperatures can delay the start of the growing season, resulting in a relatively short growing season and a narrow ring [41,42].

In addition, winter is the driest season in this monsoonal area and *Cryptomeria fortunei* favors wet conditions, thus the radial growth of these trees shows a significant positive correlation with the precipitation, relative humidity and SPEI of the previous year from November to December. Furthermore, the radial growth of *Cryptomeria fortunei* shows a significant positive correlation with different monthly SPEI of the current June to July. During the current growing season, a high summer temperature causes a high evaporative demand or lower water availability for the trees, causing stomata closure and low photosynthesis [43]. Drought stress, caused by high summer temperature, can lead to low tree growth, as evidenced in previous studies [17,19,44,45].

### 4.2. Linkage to the PDO

As shown in Figure 6, our tree-ring chronology is positively correlated with SST from previous November to current January of the previous year over the central north Pacific, and negatively correlated with SST along the western coast of North America, which resembles the negative phase of the Pacific Decadal Oscillation (PDO) [46]. PDO, as the most significant interdecadal variation signal in the middle latitude of the north Pacific, is the most important factor regulating the teleconnection between ENSO and the climate in east Asia [22,47].

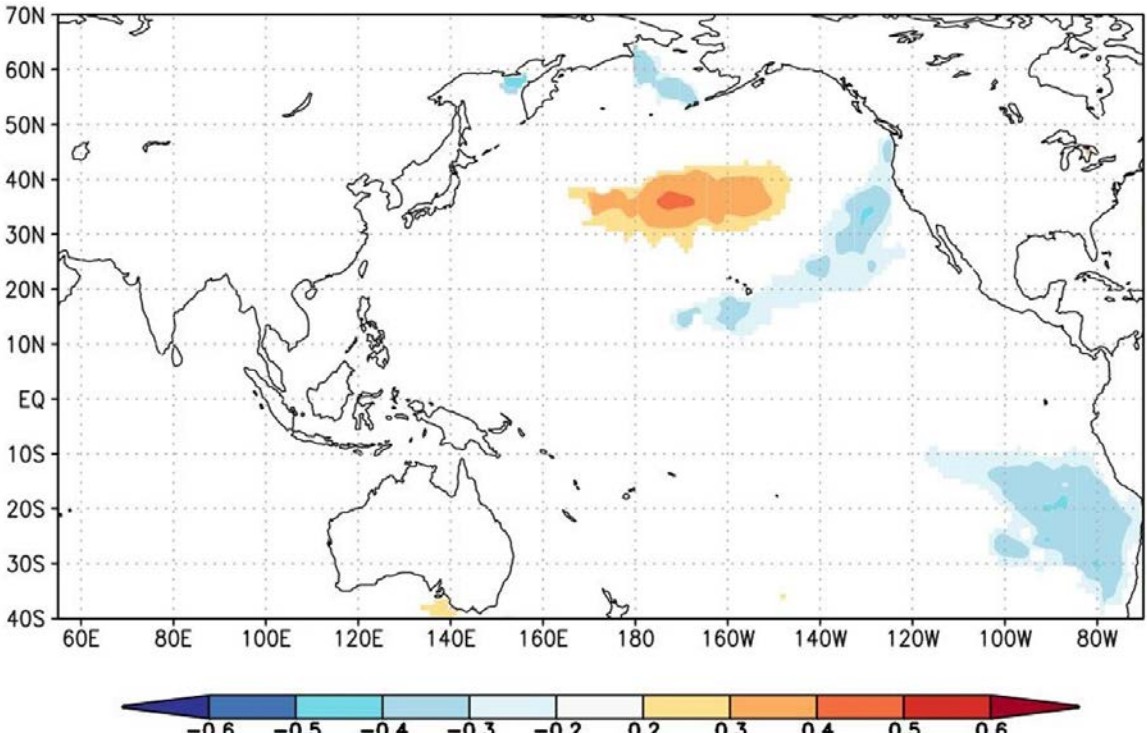

**Figure 6.** The correlation relationships between the tree-ring chronology and previous November-current January SSTs (the HadISST1 sea-surface temperature) through http://climexp.knmi.nl website during 1960–2015 ($p < 0.05$).

We compared the relationships between our tree-ring chronology, the observation PDO (http://research.jisao.washington.edu/pdo/PDO.latest) and the reconstruction PDO (MacDonald, 2005) [48], their close linkages at the decadal scale are shown in Figure 7. Our climate sensitive tree rings showed different correlations with PDO during different periods. During the 1920s–1930s and 1945–1995, the radial growth of trees was negatively correlated with observations of the PDO ($p < 0.05$). Linkages between climate sensitive tree rings and the PDO are also seen in other regions of Southeast China [18–20]. The correlation between the observation PDO and the reconstruction PDO is relatively consistent in common period (Figure 7b). The radial growth of *Cryptomeria fortunei* is negatively correlated with the PDO reconstruction before the 1900s.

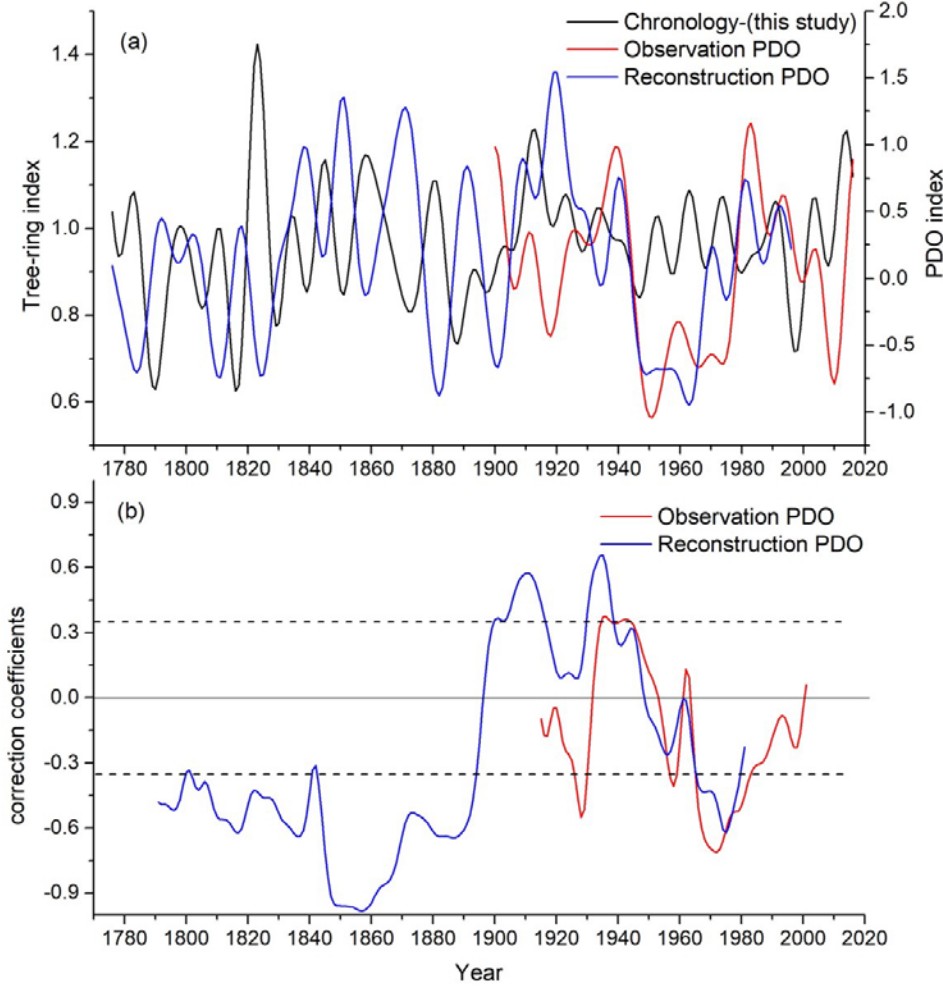

**Figure 7.** (**a**), Comparisons of tree-ring chronology (STD) and the Pacific Decadal Oscillation (PDO) smoothed using a 10-yr low-pass filter. The reconstruction PDO is from tree rings in North America [48]. (**b**), Running correlations between the tree-ring chronology and the different PDO indices based on a 31-year window. The dash lines represent the statistical significance level of 0.05. The PDO can modulate both temperature and hydroclimate in the study area, which changes in different PDO phases. Both temperature and precipitation play important roles in modulating tree growth of the study region. The PDO can affect precipitation in southeastern China by changing the intensity of EASM [49,50]. During the negative PDO phase, the EASM tends to be weakened delivering less precipitation to its marginal areas in northern China but more precipitation in its core area in southeastern China [51], which can contribute to high tree growth to the moisture limited region of our study area.

## 5. Conclusions

This study presents a tree-ring chronology of *Cryptomeria fortunei* in the Liancheng area spanning 241 years, which is currently the longest chronology in Fujian Province. Tree growth was significantly positively correlated with the precipitation and SPEI index of the previous year from November to December, as well as the temperature in winter and spring (previous December to current April). This is in line with the generally cold winter and dry summer stress growth pattern in our study region. Our tree rings also showed positive correlations with drought in previous winter and with temperatures other than summer. Our climate sensitive tree rings showed different correlations with PDO during different periods. This study provides further theoretical support for the tree-ring research and the expansion of climate information resources by using the tree-ring width of *Cryptomeria fortunei* in southeastern China. This increases the options of using longer chronologies to reconstruct the

climate in southeastern China. In addition, we should also be aware of the uncertainties about the linkages between tree ring chronology and PDO.

**Author Contributions:** Data curation, Z.Z. and C.R.; Formal analysis, D.C. and J.D.; Methodology, M.B.; Software, F.Z.; Supervision, G.Y.; Writing—original draft, Z.D.; Writing—review & editing, K.F. All authors have read and agreed to the published version of the manuscript.

**Funding:** We acknowledge the support from the Strategic Priority Research Program of the Chinese Academy of Sciences (XDB26020000), the National Science Foundation of China (41822101, 41888101, 41971022 and 41772180), fellowship for the National Youth Talent Support Program of China (Ten Thousand People Plan), fellowship for Youth Talent Support Program of Fujian Province and the innovation team project (IRTL1705).

**Conflicts of Interest:** The authors declare no conflict of interest.

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
