# Peer review of "A 241-Year Cryptomeria fortune Tree-Ring Chronology in Humid Subtropical China and Its Linkages with the Pacific Decadal Oscillation"

_atmosphere, doi:10.3390/atmos11030247_

Round 1
Reviewer 1 Report
Dong et al. submitted a manuscript entitled ‘A 241-year Cryptomeria fortunei tree-ring chronology in humid subtropical China and its linkages with the Pacific Decadel Oscillation.’
This manuscript deals certainly with an interesting topic, namely the climate history of subtropical southeastern China especially with regard to the influence of the Pacific forcing onto this area. It is, however, not a new approach. Quite a number of publications have already dealt with the problem of correlating tree-ring widths with the climate history of this area, especially the attempt to investigate the consequences of the Pacific phenomena on Southeast-China (for ex. Wang et al. (2018) Intensified variability of the el Nino-Southern Oscillation enhances its modulation on tree growth …; Chen et al. (2012) Tree-ring based winter temperature reconstruction for Changting, Fujian,….etc). Nonetheless the study goes beyond these investigations by presenting a longer tree-ring width chronology and by using the evergreen coniferous species, namely Cryptomeria fortunei.
The manuscript is reasonably well structured, however, it is somewhat superficially written and the English needs quite a number of corrections. Additionally the manuscript contains some unclarified points which are listed below and which have certainly to be rectified.
Especially the chapter ‘Materials and Methods’ shows a number of problems which have to be addressed in some detail.
Firstly, the selected trees were of rather different origin. Either selected from a natural forest specified as ‘old growth trees’ and others came from planted trees, specified as trees from ‘nearby the house, the river, and the road’ (lines 65 67). Since forest trees behave different to planted trees, the latter being presumably solitary standing trees, some arguments have to be given of how these trees behave relative to each other. How many trees were taken from the forested area and how many planted trees were used. Some arguments have to be given to justify this procedure. It is interesting to read in a later section that some unsatisfactory results are being ‘due to strong disturbances in dense forests’ (line 137).
Secondly, I think it is not justified to interpret the temperature record of Fig. 2b over the given 60 year period as a significantly increasing trend (line 63 to 64). The first half of this record is certainly not increasing and as long as further data farther back in time do not exist it is rather speculative to assume such a trend for the whole record. I only agree that the latter half of the record is increasing.
Thirdly, I disagree with the statement that ‘precipitation showed a decreasing trend’ (line 64 to 65). Optically the trend is more likely positive. In addition the linear equation given in Fig. 2c shows a positive gradient, symbolizing indeed an increasing trend!
Fourthly, if 82 cores from 30 trees were chosen, some trees must have been selected to opt for three cores and some for only two cores which is incomprehensible for me (line 81). This needs to be explained.
For the sample size I would prefer to state the number of trees, not the number of cores (Fig. 3), although this procedure is often used.
Temperature data should only be given with one digit after the point i.e. write 19.8°C instead of 19.84°C (line 59).
Since the meaning of Hakka village is not generally known it should shortly be explained (line 58).
The geographical positions of the sampling site and the meteorological stations should already be given within the text. This is also true for the other data given in Tab. 1. Table 1 is unneccessary.
The way in which the tree-ring data were treated is certainly correct and corresponds with standard dendrochronological methods.
It is amazing to me that the two meteorological stations are so closely correlated with each other, especially with regard to precipitation and humidity. On the other hand it is rather reassuring to be able to use climate data which are to the greatest possible extent reflecting the situation at the site of the sampled trees.
As far as results and discussion are concerned the arguments seem to be consistent with the available data. However, as with the previous chapter there are some corrections needed. For example Fig. 4b, d. The abscissa lists pN and pD twice, i.e. at the beginning and the end of the abscissa. The right hand side should presumably read as pDA and FS. Furthermore (line 143 to line 146) it should be noted that the given correlation is with the first order difference data. Additionally the description of correlations as mentioned in lines 155 and 156 is not compatible with Fig. 4.
Regarding the linkage between the tree-ring chronology and PDO it would also be of interest to comment on the periods from 1930 to 1945 and 1995 to 2015, respectively.
It is interesting for me to read that besides temperature and precipitation humidity has also been measured. In principle this should be a chance to include the possible effects of atmospheric vapour pressure deficit (VPD) on the growth behaviour of Cryptomeria fortunei.
VPD is a critical quantity in determining the photosynthetic activity of plants. It is known, that for example an increase in VPD rather than changes in precipitation influence the productivity. According to Yuan et al. (Sci.Adv. 2019; 5) VPD persistently increased since the late 1990s and I wonder what effect this could have had on the tree-ring record of Cryptomeria fortunei.
Reviewer 2 Report
This manuscript presents an a thoughtful, topical and timely experiment that investigates the ability to use a rather challenging tree species for a dendroclimatic analysis, from a geographical and climatic region underrepresented by such studies. The authors present a relatively extensive series of test and experiments to describe the climate signal in their trees, and even take the daring leap to make a connection between tree growth at their site with the Pacific Decadal Oscillation vis-à-vis correlations with SSTs. Though I approve of the topic, its regional relevance, the techniques used, and the well researched supporting literature, I am skeptical about the characterization of the climate signal. It is hard for me to believe the temperatures from the previous December to current August can be so significant when over much of the year the monthly correlations are no more significant than random noise and even strongly negative (ie. June)? I do believe there is a bettwe case for a significant drought signal, however the figure representing that experiment fails to be as convincing. I would suggest re-drawing figure 5 in the same manner as fig. 4, with both the un-differenced and first-differenced transformation of predictor and predictand. What is interesting, even more interesting than the teleconnection with SSTs is the purported strength of winter climate conditions on tree-growth. Here is where the experiment shows the most promise in delivering new information and insight. I would encourage the authors to just focus on that and forgo for now the considerably weaker PDO connection. However, the manuscript would benefit immensely from some careful editing to make it readable. I have made numerous suggestions and comment in the accompanying draft to more or less provide examples of this means. I would like to see this manuscript published, but definitely not in its present condition. Consequently, my recommendation is for a serious revision before acceptance.
(see: attached for more detailed comments.)

Reviewer 3 Report
Comments on Atmosphere 691045
In Eastern China, climate observation dataset only covers the past 50-60 years. Using climate proxy data and suitable methods, such as done in this manuscript, can extend the climate record to past centuries or even millennia. In addition, long-term climate records can improve our knowledge of climate change in the past and make it possible to place the current climate condition in a long-term context. In this study, the authors report the longest tree-ring width chronology covering the period 1776-2016 CE using a large number of Cryptomeria fortune tree-ring samples in Southeastern China. The manuscript has a good structure, providing important results about the relationship between tree growth and climate variability over the past decades, which will improve our understanding of future forest dynamics under the projected global warming conditions. So, I would like to suggest a minor revision with considering the below issues:
1) It is good if the authors give more analyses or discussions about the relationship between tree-ring growth and PDO. The authors show good relationship between PDO and tree-growth over the instrumental period and the 20th century, but I am interested to if such relationship is still held in preindustrial times. The long tree-ring chronology (covering the period 1776-2016 CE) provides a rather good opportunity to assess the relationship between tree growth and the PDO in the long-term perspective. So, I suggest the authors to compare the tree-ring chronology with the PDO reconstruction over their full common period.
2) More detailed information is needed for figures 6 and 7. For figure 6, the authors need to provide information about significance levels of the correlation values in figure and about the SST dataset used in calculation of correlation. For figure 7, the authors need to provide the correlation coefficients between tree-ring chronology and PDO during each focused interval; also the significance of correlation could be tested by adjusting the degrees of freedom for low-pass filtered data.
3) Grammar and expressions can be improved. For example, “A high indicates high variability between adjacent…” (Line 126); “…the correlation values between the chronology with the precipitation and humidity…” (Line 155); “…that are useful for the growth of trees the next year” (Line 179).
Round 2
Reviewer 2 Report
Dear Editor and authors,
I made final corrections and comments on these lines. These are my
final suggestions for improvement. Please do not send me another interration of the manuscript.
Sincerely,
pjk
Line#
18
47
57
62
66
67
69
121
133
134
149
160
174
181
201
202
208
210
214
217* Check to be sure this is still consistent with the figure caption
229
230
232 This 3-way analysis/ between the MacDonald recon, your chron. and PDO is confusing. First, you only introduce the reader to the MacDonald recon in the figure caption. This should be done earlier when you are introducing the reason for making the comparison in the main text.
Secondly, in the figure, the legend should also distinguish between
the three different data-types , e.g., Chronology-(this study), PDO-Hadley,
reconstruction-(MacDonald).
243
244
249
Please check the detailed corrections and comments in the attached file.

Author Response
Response : We are very grateful for the reviewer’s suggestions and comments. According to the reviewer’s comments in the accompanying draft, mistakes and suggestions have been corrected and revised in the revised manuscript.